# Rapid Evaluation and Optimization of Medium Components Governing Tryptophan Production by *Pediococcus acidilactici* TP-6 Isolated from Malaysian Food via Statistical Approaches

**DOI:** 10.3390/molecules25040779

**Published:** 2020-02-11

**Authors:** Ye Heng Lim, Hooi Ling Foo, Teck Chwen Loh, Rosfarizan Mohamad, Raha Abdul Rahim

**Affiliations:** 1Institute of Bioscience, Universiti Putra Malaysia, UPM Serdang 43400, Selangor, Malaysia; yhlim_0418@hotmail.com (Y.H.L.); farizan@upm.edu.my (R.M.); raha@utem.edu.my (R.A.R.); 2Department of Bioprocess Technology, Faculty of Biotechnology and Biomolecular Sciences, Universiti Putra Malaysia, UPM Serdang 43400, Selangor, Malaysia; 3Department of Animal Science, Faculty of Agriculture, Universiti Putra Malaysia, UPM Serdang 43400, Selangor, Malaysia; 4Institute of Tropical Agriculture and Food Security, Universiti Putra Malaysia, UPM Serdang 43400, Selangor, Malaysia; 5Institute of Tropical Forestry and Forest Products, Universiti Putra Malaysia, UPM Serdang 43400, Selangor, Malaysia; 6Department of Cell and Molecular Biology, Faculty of Biotechnology and Biomolecular Sciences, Universiti Putra Malaysia, UPM Serdang 43400, Selangor, Malaysia; 7Office of Vice Chancellor, Universiti Teknikal Malaysia Melaka, Jalan Hang Tuah Jaya, Durian Tunggal 76100, Melaka, Malaysia

**Keywords:** Tryptophan production, lactic acid bacteria, *Pediococcus acidilactici* TP-6, Plackett-Burman design, central composite design

## Abstract

Tryptophan is one of the most extensively used amino acids in livestock industry owing to its effectiveness in enhancing the growth performance of animals. Conventionally, the production of tryptophan relies heavily on genetically modified *Escherichia coli* but its pathogenicity is a great concern. Our recent study demonstrated that a lactic acid bacterium (LAB), *Pediococcus acidilactici* TP-6 that isolated from Malaysian food was a promising tryptophan producer. However, the tryptophan production must enhance further for viable industrial application. Hence, the current study evaluated the effects of medium components and optimized the medium composition for tryptophan production by *P. acidilactici* TP-6 statistically using Plackett-Burman Design, and Central Composite Design. The optimized medium containing molasses (14.06 g/L), meat extract (23.68 g/L), urea (5.56 g/L) and FeSO_4_ (0.024 g/L) significantly enhanced the tryptophan production by 150% as compared to the control de Man, Rogosa and Sharpe medium. The findings obtained in this study revealed that rapid evaluation and effective optimization of medium composition governing tryptophan production by *P. acidilactici* TP-6 were feasible via statistical approaches. Additionally, the current findings reveal the potential of utilizing LAB as a safer alternative tryptophan producer and provides insight for future exploitation of various amino acid productions by LAB.

## 1. Introduction

Fermentation medium plays an indispensable role in the industrial fermentation process due to its impact on the formation of the desired products [1]. A cost-effective medium formulation is crucial in ensuring the economic feasibility of the fermentation process. Hence, optimization of the medium composition is important in order to minimize the cost of production without compromising the production. Conventional method and statistical method are the most common methodologies employed in the optimization study. The conventional method of process optimization is also known as one-factor-at-a-time method by varying one factor while keeping the other factors unchanged until an apparent optimum condition is achieved. However, the conventional optimization method often require large number of experiments and it could be time consuming and laborious. Furthermore, this method is not suitable for multifactor optimization because it is unable to elucidate the interactions between the factors and thus incapable to detect the true optimum condition [2].

The limitation of conventional optimization method can be overcome by using statistical optimization method, which involves a collection of numerous experimental strategies, mathematical procedures and statistical inferences. Unlike conventional optimization method, the statistical optimization method is able to explain the interactions between multiple variables and determine the true optimum based on statistical approaches [3]. One of the most commonly used statistical optimization approach is response surface methodology (RSM). The first-and second-degree models are among the most frequently used approximating polynomial models in RSM. Some of the popular first-order designs that are regularly employed for optimization of the fermentation process include the Plackett-Burman Design (PBD) [4] and the Factorial Design [5]. Meanwhile, the commonly used second-order designs include Central Composite Design (CCD) [6] and Box-Behnken Design [7].

Previous optimization studies on the production of tryptophan have revolved around the conventional producer strains such as *E. coli* and *Corynebacterium glutamicum*. For instances, Faghfuri et al. [8] reported that sugar beet molasses was a good source of pyridoxal phosphate (PLP) and serine for tryptophan production by *E. coli*. Meanwhile, Cheng et al. [9] suggested that the growth and tryptophan production of *E. coli* was inhibited by acetic acid above 2 g/L and glucose concentration should be controlled at low level for tryptophan production. Moreover, Hagino and Nakayama [10] reported that molasses, casein enzymatic hydrolysate and (NH_4_)_2_SO_4_ were the best carbon, organic nitrogen and inorganic nitrogen source for tryptophan production by *C. glutamicum.* However, the use of genetically engineered and pathogenic microorganism is a major concern and has urged an exploration for a safer producer. Lactic acid bacteria (LAB) have been revealed to possess the ability to produce various amino acid in several recent studies [11,12]. Apart from its versatility in amino acid production, LAB postbiotic metabolites have been extensively reported to confer various health benefits to animals and enhance their growth performance by regulating the gastrointestinal health and immune response of the animals [13,14,15,16].

Tryptophan has gained tremendous attention in recent years, particularly in medical, feed and livestock industries. Tryptophan is known as the fourth limiting amino acid in livestock feed, right after lysine, methionine and threonine amino acids [17]. It has been demonstrated to affect both growth and neurotransmitter metabolism of poultry [18]. It also affects glucose metabolism by inhibiting gluconeogenesis [19]. Moreover, L-tryptophan was found to play a crucial role in improving the growth performance, meat quality, reducing stress, regulating insulin response and protein synthesis in muscles of pigs [20], as well as improving the feed conversion and carcass yield of broilers [21]. In medical field, tryptophan is often used as sedative and antidepressant and hence it is frequently prescribed for the treatment of schizophrenia and alcoholism [22]. Tryptophan also acts as a precursor for serotonin biosynthesis, a neurotransmitter that is responsible to relieve anxiety [23].

The effects of growth medium components on amino acid productions by LAB have not been elucidated previously, despite the effects of the M-17 medium [24] and de Man, Rogosa and Sharpe (MRS) medium [11,25] being reported, for the production of amino acid by LAB. *Pediococcus acidilactici* TP-6 was previously identified as a superior producer of tryptophan in MRS medium [25]. Nevertheless, limited knowledge regarding the nutritional requirements of *P. acidilactici* TP-6 for its growth and tryptophan production was available. Thus, the objectives of this study were to evaluate the effects of medium components on the growth and tryptophan production of *P. acidilactici* TP-6 by using PBD, followed by optimization of the medium components for tryptophan production by using steepest ascent method and CCD approaches.

## 2. Results and Discussion

### 2.1. Plackett-Burman Design

The nutritional requirement of *P. acidilactici* TP-6 for tryptophan production was studied by using PBD, where each variable was represented at two levels. A dummy variable (X) that serves as an indicator for the presence of significant interactions between the variables was incorporated in the PBD. The presence of significant interactions between the variables was indicated by high effect values of the dummy variable [26]. Table 1 shows the tryptophan production and cell population of respective trial of PBD. In general, tryptophan production was not detected in most of the experimental runs, suggesting the stringent nutrient requirement of the producer strain for tryptophan production. Run 15 recorded the highest amount of tryptophan production of 22.9 mg/L of tryptophan, followed by run 18 with 21.27 mg/L of tryptophan. However, there was no significant difference (*p* > 0.05) between the net tryptophan produced achieved in run 15 and run 18 respectively. Nevertheless, the production of tryptophan by *P. acidilactici* TP-6 in the control MRS medium (29.41 mg/L) was still significantly higher (*p* < 0.05) than those achieved in the PBD. Thus, further optimization of the medium composition is mandatory to increase tryptophan yield by *P. acidilactici* TP-6.

Table 2 presents the ANOVA of the PBD for the effects of medium components on tryptophan production by *P. acidilactici* TP−6. The low p-value of the model (0.0019) revealed that the model was highly significant (*p* < 0.01) and it is highly unlikely (>99% confidence) that the large F-value of the model was attributed to noise. Moreover, the model was able to elucidate 99% of variation in response, owing to its high coefficient of determination (R^2^ = 0.9986). Additionally, the “predicted R^2^” (0.9490) and the “adjusted R^2^” (0.9919) values were close to 1 and in reasonable agreement, implying the great correlation between experimental and predicted values and hence the suggested PBD model was a good model [27]. Furthermore, the model is suitable for navigating the design space owing to its adequate signal to noise ratio, which was reflected by the high adequate precision value of 110.181 (much greater than 4).

Based on the ANOVA of tryptophan production (Table 2), majority of the studied variables contributed significantly (*p* < 0.05) to tryptophan production by *P. acidilactici* TP-6, whereby up to 17 variables showed a p-value less than 0.05 except fructose, Tween 80 and biotin, which were not significant (*p* > 0.05). Among the 17 significant variables, 16 of them were highly significant (*p* < 0.01) except glucose and MgSO_4_, which were significant (*p* < 0.05). Additionally, the dummy variable was revealed to be highly significant (*p* < 0.01). The unexpectedly high significant effect of the dummy variable implied the presence of significant interactions between the variables [26]. Hence, a design with higher resolution is required to elucidate the interaction [28]. The net tryptophan production (Y) by *P. acidilactici* TP-6 can be expressed in terms of coded symbols (A–X) as shown in the following regression Equation (1):Y = 1.21 − 0.51A − 4.25B − 0.16C + 2.23D − 2.03E − 2.56F − 1.71G + 4.97H + 1.41J + 1.98L − 1.95M − 1.51N − 1.79O − 2.34P+ 0.56Q + 2.01R + 0.29S + 2.81T + 1.97V + 0.30W + 1.64X(1)

Figure 1 illustrates the impact of each medium component on tryptophan production by *P. acidilactici* TP-6. Among the 22 studied variables, 10 of them, including meat extract, FeSO_4_, lactose, MnSO_4_, urea, CuSO_4_, K_2_HPO_4_, MgSO_4_, biotin and Tween 80, exerted a stimulatory effect on tryptophan production, whereas the remaining 12 variables demonstrated an inhibitory effect. Out of the 10 variables that exhibited positive effect, 7 of them including meat extract, FeSO_4_, lactose, MnSO_4_, urea, CuSO_4_ and K_2_HPO_4_ were highly significant at p-value less than 0.01, whereas MgSO_4_ was significant (*p* < 0.05) as shown in Table 2. In contrast, biotin and Tween 80 did not contribute significantly (*p* > 0.05) to the tryptophan production by *P. acidilactici* TP-6. On the other hand, 8 out of the 12 variables that exhibited a negative effect were highly significant (*p* < 0.01) and one of them was significant (*p* < 0.05). Meanwhile, the other 3 variables did not affect tryptophan production significantly (*p* > 0.05).

Carbon sources play an essential role in the biosynthesis of tryptophan where the metabolism of carbon sources provides erythrose-4-phosphate and phosphoenolpyruvate, which act as precursors for tryptophan biosynthesis [29]. Among the five studied carbon sources, only lactose demonstrated a highly significant (*p* < 0.01) stimulatory effect on tryptophan production by *P. acidilactici* TP-6. Production of tryptophan by *Lactobacilli* strains in medium containing lactose as sole carbon source was also reported by Tarek and Hesham [24]. Meanwhile, the other 4 carbon sources (sucrose, molasses, glucose and fructose) exhibited a negative effect on tryptophan production by *P. acidilactici* TP-6 with sucrose, molasses and glucose being significant (*p* < 0.05) and fructose was insignificant (*p* > 0.05). This is in contrast with the findings of several other studies conducted by other researchers, where the tested carbon sources often possessed a stimulatory effect on the production of amino acid. For instance, molasses was used for the production of tryptophan by *C. glutamicum* [8]. Furthermore, molasses was reported as the most suitable carbon source for the synthesis of tryptophan by *C. glutamicum* [10]. The use of glucose for the production of various amino acid by *C. glutamicum* [30,31,32] and *E. coli* [33,34] was also well documented. Furthermore, sucrose was identified as the best carbon source for threonine production by *Escherichia coli* TRFC [35] and it was used as the sole carbon source for threonine production by a recombinant *E. coli* in a separate study conducted by Wang et al. [36]. Nevertheless, fructose was revealed as one of the best carbon sources for glutamate production by *Arthrobacter globiformis* [37]. The discrepancy of the effects of different carbon sources on amino acid production might be attributed to the use of different microorganisms as producer strains.

Among the three tested organic nitrogen sources, only meat extract demonstrated a highly significant stimulatory effect (*p* < 0.01), whereas yeast extract and peptone showed a highly significant inhibitory effect (*p* < 0.01). The organic nitrogen source is crucial for the biosynthesis of tryptophan, where it is responsible for supplying serine for the formation of tryptophan from (3-Indoyl)-glycerolphosphate catalyzed by the enzyme tryptophan synthase [38]. The high stimulatory effect of meat extract on tryptophan production by *P. acidilactici* TP-6 might be due to the rich serine content of meat extract [39]. Additionally, the positive effect of meat extract on tryptophan production could be attributed to its rich vitamin content, which acted as coenzymes for the activation of enzymes involved in the biosynthesis of tryptophan. For instance, meat extract contains an abundant amount of riboflavin [40], which can be converted into flavin mononucleotide, an essential coenzyme for chorismate synthase enzyme that is responsible for the synthesis of chorismate from 5-o-(1-carboxyvinyl)-3-phosphate [41]. Moreover, vitamin B_3,_ which is abundant in meat extract, will be metabolized into NADPH that functions as coenzyme for the action of shikimate dehydrogenase enzyme, which is accountable for the transformation of 3-dehydroshikimate into shikimate. Lim et al. [25] had demonstrated the effect of medium containing meat extract, yeast extract and peptone for the production of amino acid by LAB. In comparison, yeast extract was used for the production of tryptophan by *E. coli* in a study conducted by Faghfuri et al. [8]. In addition, Hagino and Nakayama [10] demonstrated that yeast extract was the best organic nitrogen source for tryptophan synthesis by *C. glutamicum*. Nevertheless, the present study revealed that yeast extract demonstrated an inhibitory effect on tryptophan production by *P. acidilactici* TP-6. Differences in the preference of organic nitrogen sources might be attributed to the use of different microorganism as the amino acid producer strain in the study.

On the other hand, all the inorganic nitrogen sources used in the current study contributed significantly (*p* < 0.05) to tryptophan production by *P. acidilactici* TP-6 (Table 2). However, most of them demonstrated a negative effect, except urea which exhibited a stimulatory effect. The significant positive impact of urea on the production of tryptophan might be attributed to its function to supply ammonia for the formation of anthranilate from chorismate catalyzed by the enzyme anthranilate synthase during tryptophan biosynthesis [42]. Urea was used as an inorganic nitrogen source for glutamate production by *C. glutamicum* [43] and *Brevibacterium* sp. [44], respectively, whereas NH_4_NO_3_ was utilized as the sole inorganic nitrogen source for glutamate production by LAB in a study conducted by Zareian et al. [45]. On the other hand, the use of (NH_4_)_2_SO_4_ as inorganic nitrogen source for the production of various amino acid by *C. glutamicum* [31,32] and *E. coli* [8,33,34,35,36] were well documented.

In the meantime, 5 out of 8 mineral sources including FeSO_4_, MnSO_4_, CuSO_4_, K_2_HPO_4_ and MgSO_4_ exhibited positive effect on tryptophan production by *P. acidilactici* TP-6 significantly (*p* < 0.05), whereas NaOAc contributed significantly (*p* < 0.05) on tryptophan production in a negative manner. Furthermore, KH_2_PO_4_ and ZnSO_4_ displayed an inhibitory effect on tryptophan production by the producer strain but the effect was insignificant (*p* > 0.05). The significant effects of minerals such as FeSO_4_, MnSO_4_, CuSO_4_, K_2_HPO_4_ and MgSO_4_ for the production of various amino acid have been demonstrated for *E. coli* [35,36] and *C. glutamicum* [32,46], indicating the importance of minerals for amino acid production. Many metal ions play an essential role as cofactor that is required for catalytic activity of enzymes to ensure proper functioning of biological system [47].

Findings obtained in the current study revealed that most of the divalent cations displayed a stimulatory effect on tryptophan production except Zn^2+^. The crucial role of divalent cations such as Fe^2+^, Mn^2+^, Co^2+^ and Mg^2+^ on enzymes involved in biosynthesis of tryptophan has been well documented, where they often possessed a stimulatory effect. For instances, Zalkin and Kling [48] reported that the enzyme anthranilate synthase has an absolute requirement for Mg^2+^, while Hertel et al. [49] discovered that Fe^2+^ and Co^2+^ could be used to substitute Mg^2+^ as a cofactor for anthranilate synthase. Moreover, Widholm [50] also found that Mn^2+^ or Co^2+^ could substitute Mg^2+^ for the enzyme anthranilate synthase. The negative effect of Zn^2+^ on the production of tryptophan by *P. acidilactici* TP-6 in the present study might be due to its inhibitory effect on the enzyme anthranilate synthase as suggested by Hertel et al. [49] and Widholm [50].

In comparison, the biotin vitamin B employed in the current study demonstrated a stimulatory effect on the production of tryptophan by *P. acidilactici* TP-6, despite the effect being insignificant (*p* > 0.05). This is in contrast with previous reports, whereby biotin was often essential for the production of amino acid by microorganisms such as *E. coli* [33] and *Corynebacterium* [32]. One of the possible explanations for the insignificant impact of biotin on tryptophan production by *P. acidilactici* TP-6 might be attributed to different nutrient requirement between LAB and other microorganisms for amino acid production, implying that the nutrient requirement for amino acid production could be species dependent. Another possible explanation might be due to low requirement of vitamin biotin, where it is often required in minute amount. Hence, inclusion of molasses which contain high biotin content [51] in the medium formulation could provide sufficient biotin to the producer strain of *P. acidilactici* TP-6 that used in this study.

On the other hand, the non-ionic surfactant of Tween 80 that used in the present study displayed a stimulatory effect on the production of tryptophan but it was not significant (*p* > 0.05). To the best of our knowledge, there were no reports available regarding the role of Tween 80 in the production of amino acid thus far. Despite Tween 80 has been included in the medium formulation for glutamate production by *Brevibacterium* sp. in a study conducted by Nampoothiri and Pandey [44], its significance level was not elucidated. However, Tween 80 is well-known for its crucial role for the production of various LAB metabolites. For instance, the supplementation of Tween 80 was found to critically boost the bacteriocins production [52]. Tween 80 acts as biosurfactant which is responsible to modify the fluidity and permeability of the cell membrane of producer strain. This in turn facilitates the secretion of metabolites extracellularly [53].

The effects of various medium components on the growth of *P. acidilactici* TP-6 were elucidated by the 24 experimental trials of PBD. The corresponding cell population of *P. acidilactici* TP-6 in each experimental run is presented in Table 1. Among the 24 experimental runs, the highest cell population was detected in run 22 with 9.34 log CFU/mL and it was not significantly different (*p* > 0.05) as compared to control (9.41 log CFU/mL). In contrast, run 9 and run 24 showed the lowest cell population of merely 7.33 log CFU/mL. Absence of organic nitrogen source or carbon source in both runs, which was crucial for the growth of LAB [54] and could be the reason attributing to the low cell growth in both runs.

Table 3 displays the ANOVA of the PBD for the effects of medium components on the cell growth of *P. acidilactici* TP-6. The low p-value of the model (< 0.01) revealed that the model was highly significant (*p* < 0.01) and it is highly unlikely (>99% confidence) that the large F-value of the model this large was attributed to noise. Moreover, the model exhibits great predictive strength and was able to elucidate 99% of variation in response due to its high R^2^ value (0.9986). Additionally, the “predicted R^2^” (0.9490) and the “adjusted R^2^” (0.9919) values were in reasonable agreement (difference < 0.2), implying the great correlation between experimental and predicted values and the suggested model was significant. Furthermore, the model is suitable for navigating the design space owing to its adequate signal to noise ratio, which was reflected by the high adequate precision value (44.159) that was much greater than 4.

Among the 22 studied variables, 16 variables were revealed to exhibit a significant effect (*p* < 0.05) on the growth of *P. acidilactici* TP-6. Out of the 16 significant variables, 12 of them including glucose, sucrose, molasses, yeast extract, peptone, meat extract, (NH_4_)_2_SO_4_, (NH_4_)_2_HC_6_H_5_O_7_, NaOAc, MgSO_4_, MnSO_4_ and biotin were highly significant (*p* < 0.01). Furthermore, the p-value of the dummy variable was less than 0.01, implying the possible presence of interactions between the variables, which have to be elucidated using a higher resolution design in the subsequent experiment [28]. The growth of *P. acidilactici* TP-6 (Z) can be expressed in the term of the coded symbol as shown in the following first-order regression Equation (2):Z = 8.67 + 0.17A + 0.082B + 0.023C + 0.11E + 0.28F + 0.11G + 0.21H − 0.04J − 0.025K − 0.11N + 0.14O + 0.15P + 0.11Q + 0.1R − 0.037S + 0.031U − 0.029V − 0.11W + 0.16X(2)

The effects of medium components on the growth of *P. acidilactici* TP-6 are depicted in Figure 2. Apart from biotin, (NH_4_)_2_SO_4_, K_2_HPO_4_, Tween 80, CuSO_4_ and KH_2_PO_4_, which demonstrated an inhibitory effect, the remaining studied variables affected the cell growth of *P. acidilactici* TP-6 in a positive manner. Among the 16 positive effect variables, 11 of them were significant (*p* < 0.05), except fructose, NH_4_NO_3_, FeSO_4_, lactose and urea, which were insignificant (*p* > 0.05). In addition, the present study revealed that all the tested carbon sources exerted a stimulatory effect on the cell growth of *P. acidilactici* TP-6 with glucose giving the highest stimulatory effect, followed by molasses and sucrose (Figure 2). The stimulatory effect of glucose, molasses and sucrose were significant (*p* < 0.05). Contradictorily, the stimulatory effect of fructose and lactose on the cell growth of *P. acidilactici* TP-6 was not significant (*p* > 0.05). The strong stimulatory effect of various carbon sources on the cell growth of *P. acidilactici* TP-6 implied that the carbon source was crucial for the survival and growth of producer strain of *P. acidilactici* TP-6. Furthermore, *P. acidilactici* TP-6 was able to utilize an array of carbon sources for its growth. This is in agreement with the findings reported by several studies, whereby various LAB were demonstrated to have the capability to utilize various carbon sources for their growth [55,56,57].

Similarly, the organic nitrogen sources used in the present study including yeast extract, meat extract and peptone also demonstrated highly significant (*p* < 0.01) stimulatory effect towards the growth of *P. acidilactici* TP-6 with yeast extract exhibited the highest stimulatory effect. The significant impact of organic nitrogen sources on the cell growth of *P. acidilactici* TP-6 could be attributed to its fastidious nutrient requirements. Typically, LAB are unable to survive on inorganic nitrogen solely. They were able to thrive in medium containing organic nitrogen such as complex proteins and peptides. A similar finding was also reported by Rodrigues et al. [58], where all the studied organic nitrogen sources contributed significantly to the growth of *Lactococcus lactis* 53 in a positive manner. The organic nitrogen with the highest stimulatory effect was also found to be yeast extract. Ooi et al. [59] also reported that yeast extract was crucial for bacteriocin production by *L. plantarum* I-UL4, whereby the bacteriocin production was only detected when yeast extract was present. Similar findings were reported by Lim et al. [60], where all the organic nitrogen sources demonstrated a significant stimulatory effect (*p* < 0.05) on the growth of *Pediococcus pentosaceus* TL-3, highlighting the crucial role of organic nitrogen sources for the growth of *P. pentosaceus* TL-3.

Unlike organic nitrogen sources, the effect of inorganic nitrogen sources on the growth of *P. acidilactici* TP-6 was less prominent, whereby only 2 out of the 4 studied inorganic nitrogen sources affected the cell growth of *P. acidilactici* TP-6 significantly (*p* < 0.05) with (NH_4_)_2_HC_6_H_5_O_7_ demonstrated a highly significant stimulatory effect (*p* < 0.01), whereas (NH_4_)_2_SO_4_ exerted a highly significant inhibitory effect (*p* < 0.01). The stimulatory effect of (NH_4_)_2_HC_6_H_5_O_7_ on the growth of LAB was also reported by Hwang et al. [61], where (NH_4_)_2_HC_6_H_5_O_7_ promoted the growth of *L. plantarum* Pi06. However, Rodrigues et al. [58] reported contradictory finding, where (NH_4_)_2_HC_6_H_5_O_7_ did not exhibit significant effect on the cell growth of studied LAB. This implied that the effect of inorganic nitrogen sources on the cell growth of LAB could be strain dependent. Nevertheless, although the remaining 2 inorganic nitrogen sources (NH_4_NO_3_ and urea) wielded a positive effect but the effect was insignificant (*p* > 0.05). The insignificance of inorganic nitrogen sources on LAB growth could be due to inability of LAB to assimilate inorganic nitrogen sources [62]. Furthermore, de Carvalho et al. [63] also reported that urea was not significant (*p* > 0.05) for the cell growth of LAB, whereby supplementation of urea did not improve the LAB growth.

On the other hand, majority of the studied mineral sources, including NaOAc, MgSO_4_, MnSO_4_, ZnSO_4_ and FeSO_4_ displayed a stimulatory impact on the cell growth of *P. acidilactici* TP-6, whereas the remaining 3 mineral sources (K_2_HPO_4_, CuSO_4_ and KH_2_PO_4_) displayed a negative effect. Among the 5 mineral sources with positive effect, NaOAc, MgSO_4_ and MnSO_4_ were highly significant at p-value less than 0.01 and ZnSO_4_ was significant (*p* < 0.05), whereas FeSO_4_ did not affect the growth of *P. acidilactici* TP-6 significantly (*p* > 0.05). However, out of the 3 mineral sources that possessed a negative effect, K_2_HPO_4_ and CuSO_4_ were significant (*p* < 0.05), whereas KH_2_PO_4_ was not significant (*p* > 0.05). Acetate ions are known for its great buffering capacity in maintaining the acidity of the medium. This might be the reason for the strong stimulatory effects of NaOAc on the growth of *P. acidilactici* TP-6. Since LAB is an acidophilic microorganism, which prefer an acidic growing environment [64], the presence of acetate ion to maintain the acidic growing environment could enhance the growth of LAB [65]. In contrast, the presence of phosphate ion could potentially lead to the elevation of pH, which in turn created an alkaline environment. As a consequence, the growth of acidophiles such as LAB could be retarded [60]. This could explain the inhibitory effect of KH_2_PO_4_ and K_2_HPO_4_ towards the growth of *P. acidilactici* TP-6. On the other hand, the positive impact of Mn^2+^ on the cell growth of *P. acidilactici* TP-6 was in agreement with the results obtained by Tomas et al. [65], whereby the supplementation of MnSO_4_ in the culture medium had significantly enhanced the growth of *Lactobacillus salivarius* CRL 1328. Moreover, the growth promoting effect of other mineral ions such as Mg^2+^, Fe^2+^, Mg^2+^, Ca^2+^, Co^2+^ and Cu^2+^ on LAB was also well documented, where a two-fold increment in growth was noted [66]. Despite Foucaud et al. [66] reported that Cu^2+^ stimulated the growth of LAB, yet results obtained in the current study revealed that CuSO_4_ exhibited an inhibitory effect on the growth of *P. acidilactici* TP-6. This might be due to different requirement of metal ions among different LAB strains, implying that the requirement of metal ions could be strain dependent.

Figure 2 shows that Tween 80 affected the growth of *P. acidilactici* TP-6 significantly (*p* < 0.05) in a negative manner, which was contradictory to the findings reported by Li et al. [67], whereby Tween 80 demonstrated a stimulatory effect on the growth of LAB and other microorganisms. On a separate note, the growth of *Lactobacillus casei* YIT 9018 was not affected significantly (*p* > 0.05) by Tween 80 in the study conducted by Oh et al. [68]. As a comparison, the vitamin biotin used in this study affected the growth of *P. acidilactici* TP-6 highly significantly (*p* < 0.01) in a negative manner. Similar effect was also observed for the growth of a threonine producer, *P. pentosaceus* TL-3, where biotin affected the growth of the isolate *P. pentosaceus* TL-3 significantly in a negative manner [60]. However, a contradictory finding was reported by Tripuraneni [69], where the inclusion of biotin in the growth medium of LAB enhanced the bacterial growth. The negative effect of biotin in the current study was most probably due to low biotin requirement of the producer strain. Hence, the abundant biotin content in molasses and organic nitrogen sources was sufficient to fulfil the requirement of *P. acidilactici* TP-6 [70]. Therefore, further supplementation of biotin contributed to an antagonistic effect.

A number of 17 studied variables were found to affect the production of tryptophan by *P. acidilactici* TP-6 significantly (*p* < 0.05) in the PBD study (Table 2). However, the use of all the 17 significant variables identified in the PBD for further optimization would result in large number of experimental runs. Hence, a validation test was conducted to verify the significant effects of variables identified in the PBD on tryptophan production by *P. acidilactici* TP-6 with MRS medium served as control. The medium formulation that used for validation test was described in Section 3.3. The tryptophan production, growth and serine consumption of *P. acidilactici* TP-6 that noted in different formulated media are shown in Table 4. Medium 1 that contained all the variables identified in the PBD recorded the highest tryptophan production of 26.07 mg/L, followed by Medium 5 and Medium 4 with 25.95 mg/L and 25.89 mg/L of net tryptophan produced respectively. However, there was no significant difference (*p* > 0.05) between the net tryptophan produced in Medium 1, Medium 5 and Medium 4. Nevertheless, it is noteworthy that the net tryptophan produced in the 3 media formulations was comparable to the control (26.81 mg/L), in which they were not significantly different (*p* > 0.05). This implied that Medium 1, 4 and 5 could potentially replace control MRS medium for the production of tryptophan by *P. acidilactici* TP-6.

In terms of cell growth, the highest cell population of 8.99 log CFU/mL was detected in Medium 5, yet it was still significantly lower (*p* < 0.05) in comparison to the control (9.46 log CFU/mL). In contrast, the lowest cell growth was observed in Medium 2 (7.98 log CFU/mL) and Medium 3 (8 log CFU/mL), in which they were not significantly different (*p* > 0.05). Moreover, decreasing serine concentration was detected in all formulated media, implying that *P. acidilactici* TP-6 might be able to produce tryptophan via biosynthetic pathway by converting the precursor serine to tryptophan.

The results obtained in the validation test showed that Medium 1, 4 and 5 were potential medium for tryptophan production by *P. acidilactici* TP-6 since the highest tryptophan production was noted and there was no significant difference (*p* > 0.05) between the tryptophan yield. However, Medium 1 was excluded due to its multicomponent, leaving Medium 4 and 5 for the consideration as comparable growth medium. Both Medium 4 and 5 differed in their carbon source, whereby Medium 4 contained lactose and Medium 5 contained molasses as carbon source. Despite lactose exhibiting a positive effect, yet the high cost of lactose has rendered its preference as carbon source. Hence, Medium 5 was selected for further optimization, owing to its cost effectiveness as compared to Medium 4. Moreover, the use of molasses, which is an agricultural waste as sole carbon source offer additional advantage by upgrading the agricultural waste to produce value-added product of tryptophan.

### 2.2. Steepest Ascent Method

The vicinity of optimum concentration for each medium component in Medium 5 (Molasses, meat extract, urea and FeSO_4_) were determined through a steepest ascent experiment consisting of 10 steps of ascension. The origin of each medium components in the steepest ascent experiment was fixed based on the high level (+1) of the PBD, whereas the direction and step length of each variable was determined based on the model of PBD (Equation (3)). According to the first-order model obtained in the PBD, meat extract, urea and FeSO_4_ which exerted a positive effect were adjusted towards the direction of ascension since increasing the concentration of these variables would improve the production of tryptophan. In contrast, molasses which wielded a negative effect was adjusted towards the direction of dissension, as reducing the concentration of a negative variable would enhance the tryptophan production. Meanwhile, the largest coefficient which in this case, the meat extract was used as reference to compute the step length of urea and FeSO_4_. Hence, for every 50% increment of the meat extract concentration, the concentration of urea and FeSO_4_ were increased by 20% and 28.5% respectively. At the meantime, the concentration of molasses was reduced by 10% at each run.

The cell population, net tryptophan and serine (precursor of tryptophan) produced by *P. acidilactici* TP-6 in the steepest ascent experiment are displayed in Table 5. It was clearly evidenced that the net tryptophan produced was increasing progressively along the path of steepest ascent from the origin (27 mg/L) and achieved maximum tryptophan production in run 5 (69.05 mg/L), indicating that the net tryptophan produced was enhanced approximately 2.5 folds after the optimization by steepest ascent procedure. Nevertheless, the net tryptophan produced began to decline beyond run 5, implying that further increasing the concentration of meat extract, urea and FeSO_4_ or reducing the concentration of molasses exerted inhibitory effect on the production of tryptophan by *P. acidilactici* TP-6. Furthermore, the net tryptophan produced in run 5 was significantly higher (*p* < 0.05) than control MRS medium, inferring that the formulated medium by steepest ascent method could be used as an alternative medium for the production of tryptophan by *P. acidilactici* TP-6.

Similar trend was observed for the cell population of *P. acidilactici* TP-6, whereby the cell growth increased from the origin (9.10 log CFU/mL) and the highest cell population was detected in run 5 (9.24 log CFU/mL). Thereafter, the cell growth remained unchanged up to run 7 (9.25 log CFU/mL) and the cell population decreased slowly as the concentration of medium components increased beyond run 7. Merely 8.77 log CFU/mL of cell population was noted at run 11. However, the cell population recorded in the control MRS medium (9.37 log CFU/mL) was still significantly higher (*p* < 0.05) than the cell population detected in different media formulations in the steepest ascent experiment. Interestingly, reduced serine content was detected in all the experimental runs, implying that *P. acidilactici* TP-6 might produce tryptophan via biosynthetic pathway, whereby serine precursor was converted to tryptophan, thereby resulted in a decreasing serine content. The concentrations of different medium components of run 5 (molasses, 15.04 g/L; meat extract, 24 g/L; urea, 5.4 g/L; FeSO_4_, 0.022 g/L) were subsequently employed as the center point for further optimization by using CCD.

### 2.3. Central Composite Design

The concentrations of molasses, meat extract, urea and FeSO_4_ were further optimized by using CCD of Response Surface Methodology (RSM) after the steepest ascent procedure. The concentration of molasses, meat extract, urea and FeSO_4_ were assigned to five levels—high level (+1), low level (*−*1), central point (0) and 2 axial points (±α). Hence, a total of 30 experimental runs were suggested by the CCD and their corresponding experimental and predicted net tryptophan produced are shown in Table 6 respectively. In general, the highest net tryptophan produced was detected in runs 25–30, where all the variables were set at the center point and the respective net tryptophan produced was up to 70 mg/L, followed by run 24 (66.07 mg/L) which constituted of molasses, meat extract and urea that fixed at center point, while the FeSO_4_ was supplemented at +2 level. The net tryptophan production recorded in runs 25–30 were significantly higher (*p* < 0.05) as compared to the other experiment runs, as well as the control MRS medium (28.18 mg/L).

The data were then analyzed with different regression models to investigate which model is best fitted to describe the relation between the variables and the production of tryptophan produced by *P. acidilactici* TP−6 as presented in Table 7. Based on the ANOVA table, it is clearly evidenced that the data were best fitted to a quadratic polynomial model. Among the 4 tested polynomial models, only the quadratic model was significant (*p* < 0.05), whereas the other polynomial models were insignificant (*p* > 0.05). Additionally, the quadratic model was highly predictive owing to its exceptionally high adjusted R^2^ value (0.9837) and high predicted R^2^ value (0.9586), which was not observed for other polynomial models. Furthermore, the p-value of the lack of fit test of the quadratic model (0.2196) implied that the lack of fit was not significant (*p* > 0.05) and the model can be used to explain and predict the response, which in this case the tryptophan production. This was evidenced by the good agreement between the predicted and experimental tryptophan production as presented in Table 6. Unlike the cubic polynomial model, the presence of aliased effects between the variables was not detected in the quadratic model. Hence, the production of tryptophan by *P. acidilactici* TP−6 can be best represented by the quadratic model. The following quadratic Equation (3) elucidated the effects of molasses (A), meat extract (B), urea (C) and FeSO_4_ (D) on tryptophan production by *P. acidilactici* TP-6 (Y) in terms of coded symbols (A–D):Y = 69.42 − 1.39A − 0.29B + 1.81C + 1.61D + 0.9AB - 0.084AC − 1.00AD + 0.58BC − 0.11BD − 0.45CD − 2.58A^2^ − 3.41B^2^ − 2.88C^2^ − 1.79D^2^(3)

The statistical significance of the quadratic model and the variables were evaluated by F-test and the ANOVA is presented in Table 8. The low p-value of the model (<0.01) implied that the model was highly significant (*p* < 0.01) and it was highly unlikely (>99% confidence) that the large F-value of the model this large was attributed to noise. Moreover, the model exhibited great predictive strength and was able to explain 99.2% of variation in response due to its high R^2^ value (0.9916). Additionally, the “predicted R^2^” (0. 9586) and the “adjusted R^2^” (0.9837) values were in reasonable agreement (difference < 0.2), implying the great correlation between experimental and predicted values and the suggested model was significant. This was further supported by the insignificant lack of fit (*p* > 0.05), which was indicated by the high p-value of lack of fit test (0.22). Furthermore, the model was suitable for navigating the design space owing to its adequate signal to noise ratio which was reflected by the high adequate precision value (33.028) that was much greater than the threshold value of f4. On the other hand, the ANOVA results revealed that all the linear coefficients and quadratic coefficients affected tryptophan production significantly (*p* < 0.01) except the linear coefficient of meat extract (B). Furthermore, the interaction coefficient AB, AD, BC and BC were found to contribute significantly (*p* < 0.05) to the production of tryptophan by *P. acidilactici* TP−6.

The relationship between the coded variables and response was subsequently examined by constructing the three-dimensional surface plots as shown in Figure 3, Figure 4, Figure 5, Figure 6, Figure 7 and Figure 8. The interaction between molasses and meat extract is depicted in Figure 3 by maintaining the concentration of urea and FeSO_4_ at 5.4 g/L and 0.022 g/L, respectively, as the center point. Increased tryptophan production was noted with increased concentration of meat extract and molasses. The highest net tryptophan produced was detected when molasses and meat extract were both between the levels of −1 to +1. The synergistic effect of molasses and meat extract (AB) was highly significant as reflected by the p-value of less than 0.01 (Table 8). Figure 4 illustrates the response surface of tryptophan production with respect to molasses and urea by keeping the concentration of meat extract and FeSO_4_ at the center point (24 g/L, 0.022 g/L). Increasing concentration of both molasses and urea resulted in higher net tryptophan produced. The highest net tryptophan produced was detected when molasses was in the range of −1 to 0 and urea was between 0 to +1. However, the tryptophan production was retarded when the concentration of molasses and urea was changed beyond the above-mentioned boundaries. The p-value of the interaction effect of AC (0.63) indicated that it was not significant (Table 8).

Figure 5 illustrates the combined effect between molasses and FeSO_4_ while keeping the level of meat extract and urea at 24 g/L and 5.4 g/L, respectively, which were the center points. Similarly, increasing concentration of both molasses and FeSO_4_ enhanced the net tryptophan produced. The highest net tryptophan production was detected when molasses was at the center point and FeSO_4_ was between the center point (0) and high level (+1). ANOVA (Table 8) revealed that the interaction effect between molasses and FeSO_4_ (AD) was highly significantly (*p* < 0.01) on the net tryptophan produced. Figure 6 shows the interaction between the meat extract and urea, in which the concentration of FeSO_4_ and molasses were maintained at 0.022 g/L and 15.04 g/L, respectively, which were the center points. Increasing concentration of meat extract and urea improved the net tryptophan production by *P. acidilactici* TP-6. The highest tryptophan production was detected when both urea and meat extract were between low level (−1) and high level (+1). Increasing or decreasing the concentration of either meat extract or urea beyond these boundaries reduced the net tryptophan produced substantially. Based on Table 8, the interaction effect of meat extract and urea (BC) was highly significant (*p* < 0.01).

Figure 7 depicts the three-dimensional surface plot of net tryptophan produced as a function of meat extract and FeSO_4_ while keeping the concentration of molasses and urea at the center point (15.04 g/L, 5.4 g/L). Increasing concentration of both FeSO_4_ and meat extract elevated the net tryptophan produced. The highest production was observed when meat extract was in the range of −1 to +1 while the FeSO_4_ may vary between −1 to +2. The synergistic effect of meat extract and FeSO_4_ (BD) was insignificant (*p* > 0.05) as reflected by the high p-value (0.53) in the ANOVA analysis (Table 8). On the other hand, the combined effect of urea and FeSO_4_ is illustrated in Figure 8, in which the concentration of meat extract and molasses was kept constant at the center point (24 g/L, 15.04 g/L). Enhanced production of tryptophan was noted upon increment of the urea and FeSO_4_ concentration. The highest net tryptophan production was observed when urea was supplemented between the range of 0 to +1 and FeSO_4_ was in the range of −1 up to +2. The *p*-value of the interaction coefficient CD (0.02) revealed that it contributed significantly (*p* < 0.05) to the production of tryptophan by *P. acidilactici* TP-6.

By considering criteria such that the response is maximized and all the variables were in the range from −1 to +1, the optimum concentration of molasses (14.06 g/L), meat extract (23.68 g/L), urea (5.56 g/L) and FeSO_4_ (0.024 g/L) were revealed with a predicted tryptophan production of 70.38 mg/L. Upon validation by cultivating *P. acidilactici* TP-6 in the optimized medium proposed by the model, up to 68.05 mg/L of tryptophan production was achieved by *P. acidilactici* TP-6 experimentally. Despite the experimental tryptophan production being slightly lower than the predicted value; however, they were not significantly different (*p* > 0.05). The tryptophan production recorded by *P. acidilactici* TP-6 in the optimized medium (68.05 mg/L) was enhanced by 150% in comparison to the control MRS medium (26.45 mg/L). Meanwhile, the tryptophan productivity by the producer strain in the optimized medium (3.40 mg/L/h) was improved by approximately 2.5 folds as compared to the control MRS medium (1.32 mg/L/h). In comparison with the production of tryptophan by *L. delbrueckii* subsp. *bulgaricus* (7.4 mg/L) in a study conducted by Simova et al. [71], *P. acidilactici* TP-6 produced a more than 9 times higher amount of tryptophan by using optimized medium as noted in the current study. Moreover, less than 2 mg/L of tryptophan production was reported for *Lactobacilli* [24], which was tremendously lower than the tryptophan production reported in the present study. Thus, the findings obtained in this study revealed that a rapid evaluation and effective optimization of medium composition governing tryptophan production by *P. acidilactici* TP-6 were feasible via statistical approaches, whereby the production of tryptophan demonstrated by *P. acidilactici* TP-6 in the optimized medium was much higher than those reported in several studies using a non-optimized medium, inferring the feasibility of utilizing *P. acidilactici* TP-6 as a food-grade alternative producer for the production of tryptophan. Additionally, the current findings revealed the potential of utilizing LAB as a safer alternative tryptophan producer, as well as providing basis for the exploitation of various amino acid productions by LAB in the future.

## 3. Materials and Methods

### 3.1. Inoculum Preparation

*P. acidilactici* TP-6 that was previously isolated from Malaysian fermented food, *Tempeh* [72], was employed as the producer strain for tryptophan production in this study. The strain was cultivated and preserved as described by Kareem et al. [73]. The inoculum preparation was performed as described by Lim et al. [60].

### 3.2. Experimental Design

PBD was first employed to elucidate the significance of medium components on the production of tryptophan by *P. acidilactici* TP-6, followed by validation of the effects of significant variables that were most crucial for tryptophan production by *P. acidilactici* TP-6. The range of optimum concentration for each significant varieble was subsequently estimated by using steepest ascent method, followed by the optimization of the concentrations of medium components for the production of tryptophan by *P. acidilactici* TP-6 via CCD of RSM.

### 3.3. Plackett-Burman Design

The nutritional requirement of *P. acidilactici* TP-6 for tryptophan production was evaluated by using PBD [74]. The design of experiment and statistical analysis of data were performed by using Design Expert statistical software version 9.0.6.2 (State-Ease Inc, Minneapolis, MN, USA). A number of 22 medium components which might affect amino acid production were selected based on published reports and control MRS medium composition and evaluated in the current study by assigning each variable at two distinct levels, namely low level (−1) and high level (+1) as presented in Table 9 [60].

Table 10 presents matrixes of the PBD constituting of 24 experimental runs as suggested by the software.

The following first-order model was used to express the response of the PBD:(4)$$Y=\beta_{0}+\sum_{i=1}^{22}\beta {\;}_{i}X_{i},$$
where:

*Y* = Response variable 

*β*_0_ = Interception coefficient

*β_i_* = Coefficients of linear effects of the independent variables (*X*_1_ − *X*_22_)

The effects of the significant variables identified in the PBD was subsequently validated by formulating five different media as shown in Table 11.

Formulation 1 constituted of all the significant variables; Formulation 2 was made up of only the significant variables with positive effect; Formulation 3 was comprised all the variables with stimulatory effects irrespective of their significance level; Formulation 4 contained 4 main components, which represent the carbon source, organic nitrogen source, inorganic nitrogen source and mineral source with the highest positive effect; Formulation 5 constituted of the similar medium composition as Formulation 4 except lactose was replaced with molasses, which exhibited a strong stimulatory effect on the growth and the cost was the lowest among the five carbon sources.

### 3.4. Steepest Ascent Method

The neighborhood of the optimum concentrations of each significant variable (molasses, meat extract, urea and FeSO_4_) was subsequently estimated by using the steepest ascent method. The direction of ascent or descent of each variable was determined by using the first-order model from PBD as a guideline, where variables bearing a positive sign was moved along the path of Steepest Ascent and vice versa. Meanwhile, the coefficient with the highest value, in this case meat extract was used as a benchmark to compute the step length of urea and FeSO_4_. Table 12 shows the Steepest Ascent design consisting of 10 steps that governed the production of tryptophan by *P. acidilactici* TP-6. The concentration of urea and FeSO_4_ was increased by 20% and 28.5% respectively for every 50% increment of the meat extract concentration. Meanwhile, the molasses concentration was reduced by 10% at each run.

### 3.5. Central Composite Design

Subsequently, the CCD was employed for the determination of optimum concentration of molasses, meat extract, urea and FeSO_4_ that required for tryptophan production by *P. acidilactici* TP-6. Design Expert statistical software version 9.0.6.2 (State-Ease Inc, Minneapolis, MN, USA) was used for designing of experiment and statistical analysis. Table 13 presents the details of each medium component used in the CCD, where each variable was assigned to five distinct levels (−α, −1, 0, +1, +α). The axial distance selected in this study was two, such that the design was rotatable.

Table 14 shows a total of 30 experimental runs suggested by CCD software, comprising of 16 factorial points, 8 axial points and 6 central points. The following second-order model could be used to express the relationship of the response with each variable:(5)$$Y=\beta_{0}+\sum \beta_{j}X_{j}+\sum \beta_{j^{2}}X_{j^{2}}+\sum \beta_{jk}X_{j}X_{k}$$
where:

*Y* = Response variable

*ß*_0_ = Interception coefficient

*ß*_j_ = Linear coefficients

*ß*_j_^2^ = Quadratic coefficients

*ß*_jk_ = Interactive coefficients

### 3.6. Production of Tryptophan

Cultivation of the bacterial strain for tryptophan production was performed by inoculating 10% (*v*/*v*) of inoculum into the media, followed by incubation for 20 h at 30 °C [25].

### 3.7. Analytical Methods

The tryptophan content of the cultured broth was determined after separation of biomass by centrifugation for 10 min at 10,000× *g*, 4 °C. Meanwhile, the cell growth was determined by using the biomass. The determination of cell population and tryptophan content was conducted as described by Lim et al. [60].

## 4. Conclusions

Seventeen of the 22 studied variables were found to exhibit significant effects (*p* < 0.05) on the production of tryptophan by *P. acidilactici* TP-6, while the remaining 5 variables including fructose, KH_2_PO_4_, Tween 80, ZnSO_4_ and biotin did not affect the net tryptophan produced significantly (*p* > 0.05). On the other hand, s16 variables contributed significantly (*p* < 0.05) to the growth of *P. acidilactici* TP-6, except for fructose, lactose, KH_2_PO_4_, urea, NH_4_NO_3_ and FeSO_4_, which had no significant effect (*p* > 0.05) on the cell growth of *P. acidilactici* TP-6. A medium constituting of molasses, meat extract, urea and FeSO_4_ was proven to be the best medium for the production of tryptophan by *P. acidilactici* TP-6, where it permitted the highest amount of tryptophan production (25.95 mg/L) with the additional advantage of cost competitiveness. Furthermore, the tryptophan production was comparable to the control MRS medium with no significant difference (*p* > 0.05). Hence, molasses, meat extract, urea and FeSO_4_ were subsequently selected for further optimization. The application of the Steepest Ascent procedure has successfully improved the net tryptophan produced by approximately 2.5 folds, from 27 mg/L at run 1 to 69.05 mg/L at run 5. Subsequently, the optimum concentration of each medium components was determined by using CCD and it was suggested that the highest predicted tryptophan production (70.38 mg/L) could be achieved by using the combination of molasses (14.06 g/L), meat extract (23.68 g/L), urea (5.56 g/L) and FeSO_4_ (0.024 g/L). An amount of 68.05 mg/L of tryptophan was produced by *P. acidilactici* TP-6 upon validation by cultivating the producer strain in the optimized medium. There was no significant difference (*p* > 0.05) between the tryptophan production predicted by the model with the experimental tryptophan production. Up to 150% enhancement of tryptophan production by *P. acidilactici* TP-6 was achieved by using the optimized medium, whereas the cost of the fermentation medium was reduced by 11% as compared to the control MRS medium.

## Figures and Tables

**Figure 1 molecules-25-00779-f001:**
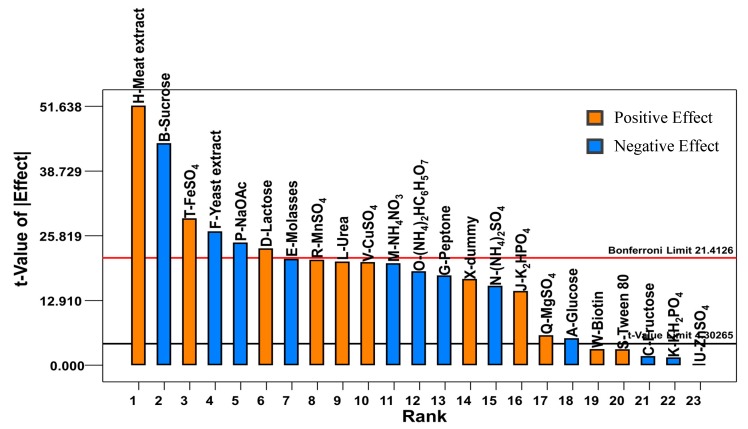
Effects of different medium components on tryptophan production by *P. acidilactici* TP-6.

**Figure 2 molecules-25-00779-f002:**
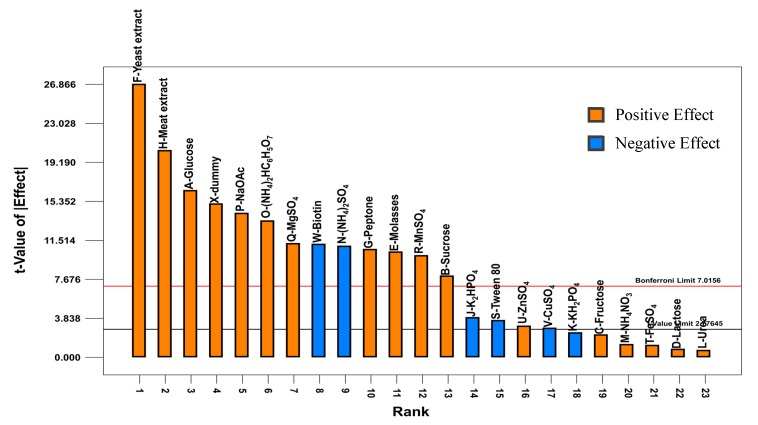
Effects of medium compositions on growth of *P. acidilactici* TP-6.

**Figure 3 molecules-25-00779-f003:**
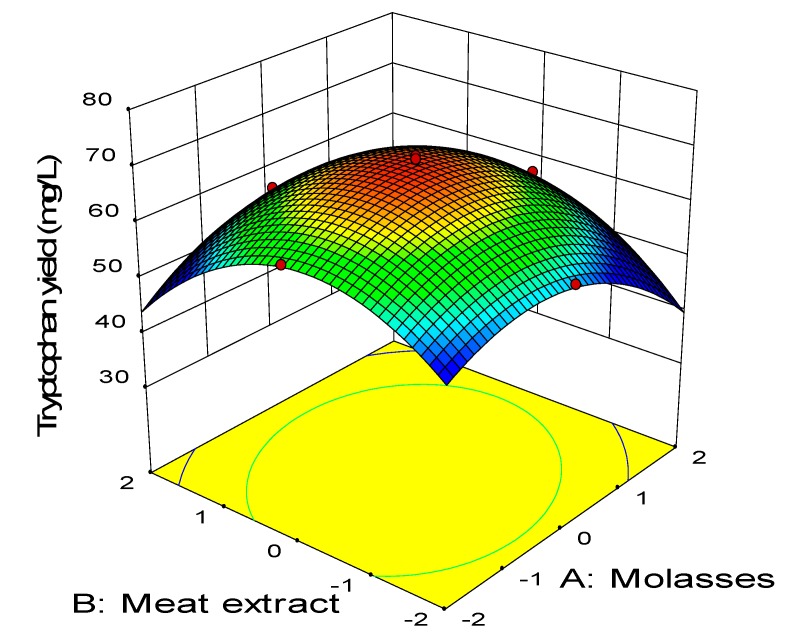
Response surface of tryptophan production by *P. acidilactici* TP-6 as a function of molasses and meat extract.

**Figure 4 molecules-25-00779-f004:**
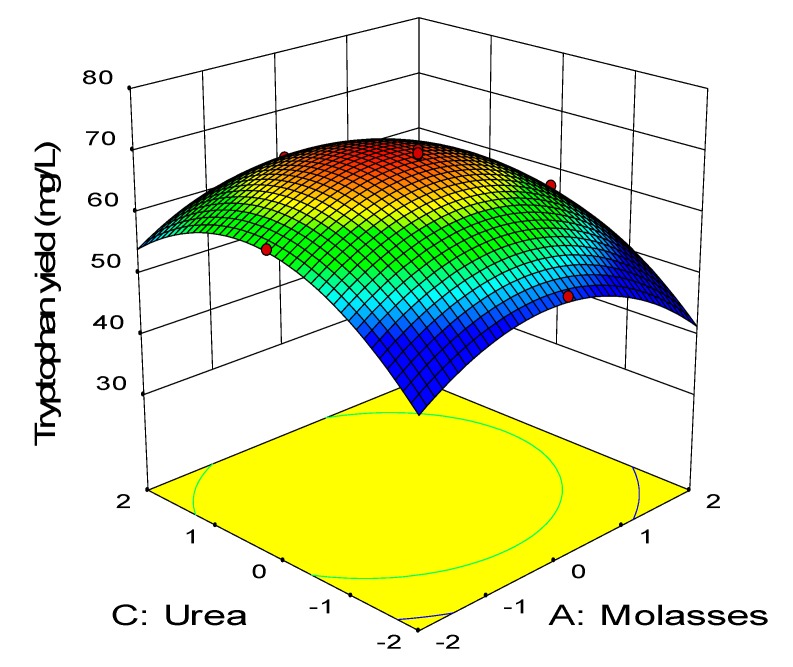
Response surface of tryptophan production by *P. acidilactici* TP-6 as a function of molasses and urea.

**Figure 5 molecules-25-00779-f005:**
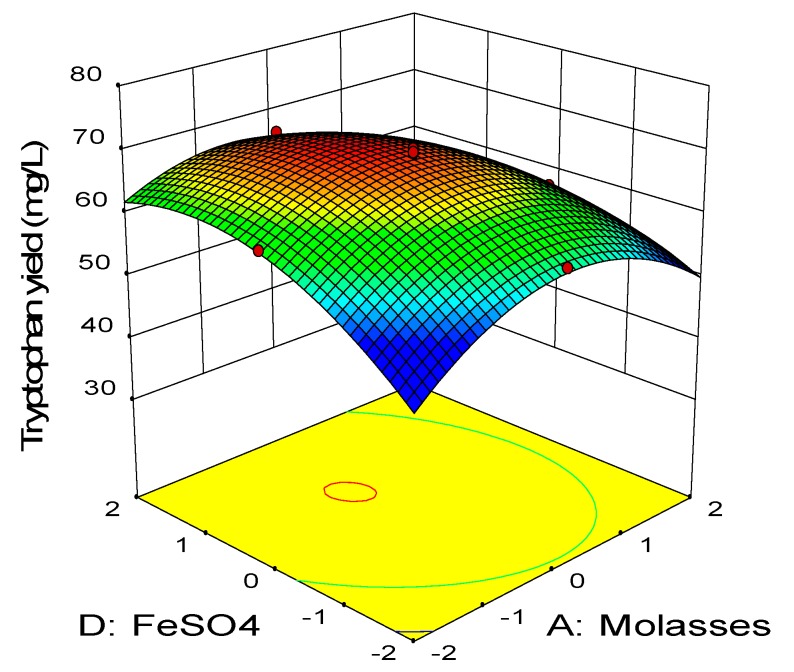
Response surface of tryptophan production by *P. acidilactici* TP-6 as a function of molasses and FeSO_4_.

**Figure 6 molecules-25-00779-f006:**
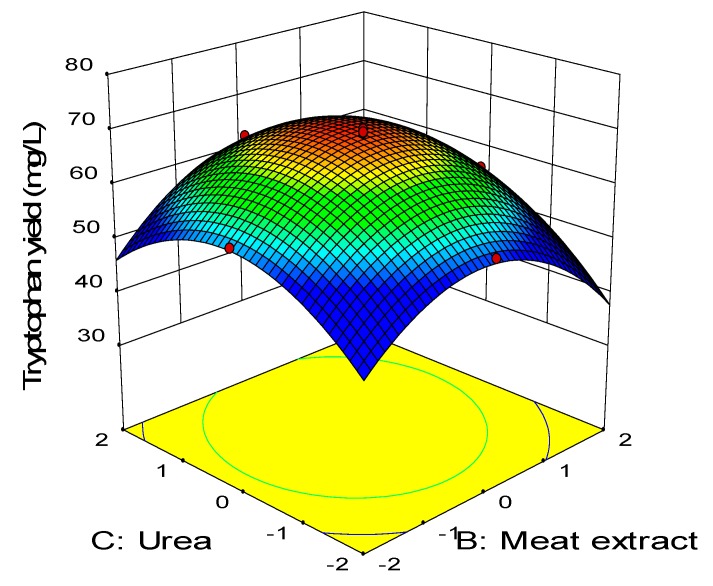
Response surface of tryptophan production by *P. acidilactici* TP-6 as a function of meat extract and urea.

**Figure 7 molecules-25-00779-f007:**
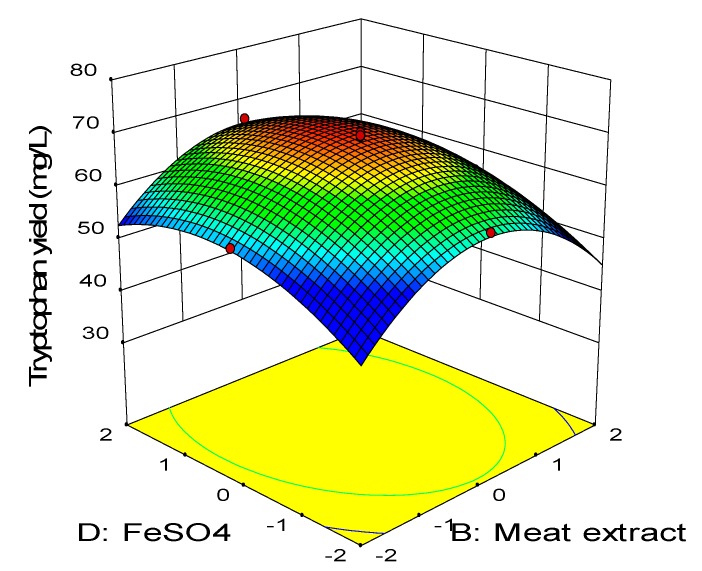
Response surface of tryptophan production by *P. acidilactici* TP-6 as a function of meat extract and FeSO_4._

**Figure 8 molecules-25-00779-f008:**
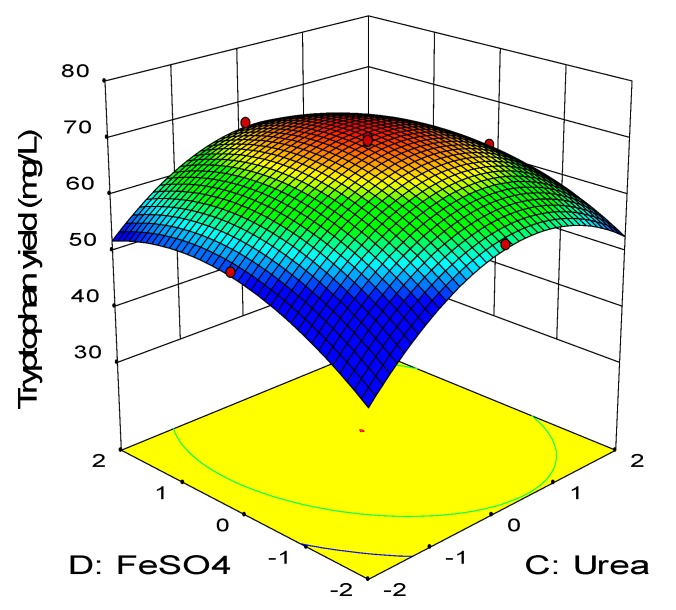
Response surface of tryptophan production by *P. acidilactici* TP-6 as a function of urea and FeSO_4._

**Table 1 molecules-25-00779-t001:** Plackett-Burman Design (PBD) matrixes for 22 variables with coded values and their corresponding tryptophan production and cell population of *P. acidilactici* TP-6.

Std Run	A	B	C	D	E	F	G	H	J	K	L	M	N	O	P	Q	R	S	T	U	V	W	X	Tryptophan Production (mg/L)	Cell Population (logCFU/mL)
1	1	1	1	1	1	-1	1	−1	1	1	−1	−1	1	1	−1	−1	1	−1	1	−1	−1	−1	−1	0.00 ± 0.00 ^H^	8.47 ± 0.04 ^I^
2	−1	1	1	1	1	1	−1	1	−1	1	1	−1	−1	1	1	−1	−1	1	−1	1	−1	−1	−1	0.00 ± 0.00 ^H^	9.26 ± 0.01 ^B^
3	−1	−1	1	1	1	1	1	−1	1	−1	1	1	−1	−1	1	1	−1	−1	1	−1	1	−1	−1	0.42 ± 0.05 ^G^	8.82 ± 0.03 ^F^
4	−1	−1	−1	1	1	1	1	1	−1	1	−1	1	1	−1	−1	1	1	−1	−1	1	−1	1	−1	0.34 ± 0.02 ^G^	8.77 ± 0.04 ^F^
5	−1	−1	−1	−1	1	1	1	1	1	−1	1	−1	1	1	−1	−1	1	1	−1	−1	1	−1	1	9.35 ± 0.05 ^D^	9.10 ± 0.02 ^D^
6	1	−1	−1	−1	−1	1	1	1	1	1	−1	1	−1	1	1	−1	−1	1	1	−1	−1	1	−1	0.00 ± 0.00 ^H^	9.04 ± 0.01 ^E^
7	−1	1	−1	−1	−1	−1	1	1	1	1	1	−1	1	−1	1	1	−1	−1	1	1	−1	−1	1	6.81 ± 0.07 ^E^	8.70 ± 0.01 ^G^
8	1	−1	1	−1	−1	−1	−1	1	1	1	1	1	−1	1	−1	1	1	−1	−1	1	1	−1	−1	16.69 ± 0.51 ^C^	8.75 ± 0.02 ^FG^
9	−1	1	−1	1	−1	−1	−1	−1	1	1	1	1	1	−1	1	−1	1	1	−1	−1	1	1	−1	0.00 ± 0.00 ^H^	7.33 ± 0.01 ^M^
10	−1	−1	1	−1	1	−1	−1	−1	−1	1	1	1	1	1	−1	1	−1	1	1	−1	−1	1	1	0.00 ± 0.00 ^H^	7.89 ± 0.02 ^L^
11	1	−1	−1	1	−1	1	−1	−1	−1	−1	1	1	1	1	1	−1	1	−1	1	1	−1	−1	1	0.00 ± 0.00 ^H^	9.22 ± 0.02 ^BC^
12	1	1	−1	−1	1	−1	1	−1	−1	−1	−1	1	1	1	1	1	−1	1	−1	1	1	−1	−1	0.00 ± 0.00 ^H^	8.79 ± 0.02 ^F^
13	−1	1	1	−1	−1	1	−1	1	−1	−1	−1	−1	1	1	1	1	1	−1	1	−1	1	1	−1	0.00 ± 0.00 ^H^	9.02 ± 0.03 ^E^
14	−1	−1	1	1	−1	−1	1	−1	1	−1	−1	−1	−1	1	1	1	1	1	−1	1	−1	1	1	4.85 ± 0.04 ^F^	8.60 ± 0.00 ^H^
15	1	−1	−1	1	1	−1	−1	1	−1	1	−1	−1	−1	−1	1	1	1	1	1	−1	1	−1	1	22.94 ± 0.79 ^B^	9.19 ± 0.01 ^C^
16	1	1	−1	−1	1	1	−1	−1	1	−1	1	−1	−1	−1	−1	1	1	1	1	1	−1	1	−1	0.00 ± 0.00 ^H^	8.75 ± 0.03 ^FG^
17	−1	1	1	−1	−1	1	1	−1	−1	1	−1	1	−1	−1	−1	−1	1	1	1	1	1	−1	1	0.00 ± 0.00 ^H^	8.75 ± 0.02 ^FG^
18	1	−1	1	1	−1	−1	1	1	−1	−1	1	−1	1	−1	−1	−1	−1	1	1	1	1	1	−1	21.27 ± 0.24 ^B^	7.88 ± 0.02 ^L^
19	−1	1	−1	1	1	−1	−1	1	1	−1	−1	1	−1	1	−1	−1	−1	−1	1	1	1	1	1	10.58 ± 0.25 ^D^	8.47 ± 0.01 ^K^
20	1	−1	1	−1	1	1	−1	−1	1	1	−1	−1	1	−1	1	−1	−1	−1	−1	1	1	1	1	0.00 ± 0.00 ^H^	8.50 ± 0.01 ^IJ^
21	1	1	−1	1	−1	1	1	−1	−1	1	1	−1	−1	1	−1	1	−1	−1	−1	−1	1	1	1	0.00 ± 0.00 ^H^	9.11 ± 0.01 ^D^
22	1	1	1	−1	1	−1	1	1	−1	−1	1	1	−1	−1	1	−1	1	−1	−1	−1	−1	1	1	0.00 ± 0.00 ^H^	9.34 ± 0.02 ^AB^
23	1	1	1	1	−1	1	−1	1	1	−1	−1	1	1	−1	−1	1	−1	1	−1	−1	−1	−1	1	0.35 ± 0.01 ^G^	9.05 ± 0.01 ^E^
24	−1	−1	−1	−1	−1	−1	−1	−1	−1	−1	−1	−1	−1	−1	−1	−1	−1	−1	−1	−1	−1	−1	−1	0.00 ± 0.00 ^H^	7.33 ± 0.02 ^M^
MRS	1	−1	−1	−1	−1	1	1	1	1	−1	−1	−1	−1	1	1	1	1	1	−1	−1	−1	−1	−1	29.41 ± 0.76 ^A^	9.41 ± 0.01 ^A^

Note: Values are mean ± SEM, *n* =3. Mean ± SEM within the same column that share a common superscript (A–M) are not significantly different (*p* > 0.05).

**Table 2 molecules-25-00779-t002:** ANOVA of PBD for the effects of medium components on tryptophan production by *P. acidilactici* TP-6.

Source	Sum of Squares	df	Mean Square	F Value	*p*-Value Prob > F	
Model	2428.91	21	115.66	520.43	<0.01	significant
A-Glucose	6.13	1	6.13	27.56	0.03	significant
B-Sucrose	433.36	1	433.36	1949.90	<0.01	significant
C-Fructose	0.61	1	0.61	2.76	0.24	
D-Lactose	119.36	1	119.36	537.05	<0.01	significant
E-Molasses	98.68	1	98.68	444.03	<0.01	significant
F-Yeast extract	156.85	1	156.85	705.75	<0.01	significant
G-Peptone	70.03	1	70.03	315.12	<0.01	significant
H-Meat extract	592.62	1	592.62	2666.50	<0.01	significant
J-K_2_HPO_4_	47.76	1	47.76	214.88	<0.01	significant
L-Urea	93.75	1	93.75	421.83	<0.01	significant
M-NH_4_NO_3_	90.83	1	90.83	408.69	<0.01	significant
N-(NH_4_)_2_SO_4_	54.88	1	54.88	246.92	<0.01	significant
O-(NH_4_)_2_HC_6_H_5_O_7_	77.00	1	77.00	346.49	<0.01	significant
P-NaOAc	131.64	1	131.64	592.30	<0.01	significant
Q-MgSO_4_	7.54	1	7.54	33.93	0.03	significant
R-MnSO_4_	97.21	1	97.21	437.39	<0.01	significant
S-Tween 80	2.04	1	2.04	9.19	0.09	
T-FeSO_4_	188.94	1	188.94	850.15	<0.01	significant
V-CuSO_4_	92.90	1	92.90	418.02	<0.01	significant
W-Biotin	2.09	1	2.09	9.42	0.09	
X-dummy	64.69	1	64.69	291.09	<0.01	significant
Residual	0.44	2	0.22			
Cor Total	2429.36	23				

Note: R^2^: 0.9998; Adj R^2^: 0.9979; Pred R^2^: 0.9737; Adeq Precision: 110.181.

**Table 3 molecules-25-00779-t003:** ANOVA of PBD for the effects of medium components on the growth of *P. acidilactici* TP-6.

Source	Sum of Squares	df	Mean Square	F Value	*p*-Value Prob > F	
Model	7.15	19	0.38	148.54	<0.01	significant
A-Glucose	0.68	1	0.68	269.27	<0.01	significant
B-Sucrose	0.16	1	0.16	63.98	<0.01	significant
C-Fructose	0.01	1	0.01	4.88	0.09	
E-Molasses	0.27	1	0.27	107.47	<0.01	significant
F-Yeast extract	1.83	1	1.83	721.80	<0.01	significant
G-Peptone	0.29	1	0.29	112.71	<0.01	significant
H-Meat extract	1.05	1	1.05	414.04	<0.01	significant
J-K_2_HPO_4_	0.04	1	0.04	15.23	0.02	significant
K-KH_2_PO_4_	0.01	1	0.01	5.82	0.07	
N-(NH_4_)_2_SO_4_	0.30	1	0.30	119.34	<0.01	significant
O-(NH_4_)_2_HC_6_H_5_O_7_	0.46	1	0.46	179.90	<0.01	significant
P-NaOAc	0.51	1	0.51	200.63	<0.01	significant
Q-MgSO_4_	0.32	1	0.32	125.26	<0.01	significant
R-MnSO_4_	0.25	1	0.25	100.07	<0.01	significant
S-Tween 80	0.03	1	0.03	13.04	0.02	significant
U-ZnSO_4_	0.02	1	0.02	9.40	0.04	significant
V-CuSO_4_	0.02	1	0.02	8.15	0.05	significant
W-Biotin	0.31	1	0.31	123.62	<0.01	significant
X-dummy	0.58	1	0.58	227.63	<0.01	significant
Residual	0.01	4	0.00			
Cor Total	7.16	23				

Note: R^2^: 0.9986; Adj R^2^: 0.9919; Pred R^2^: 0.9490; Adeq Precision: 44.159.

**Table 4 molecules-25-00779-t004:** Growth, net tryptophan and serine produced by *P. acidilactici* TP-6 in formulated media.

Media	Cell Population (Log CFU/mL)	Tryptophan Production (mg/L)	Serine Consumption (mg/L)
**1**	8.95 ± 0.01 ^B^	26.07 ± 0.86 ^AB^	9.95 ± 0.43 ^A^
**2**	7.98 ± 0.01 ^D^	25.16 ± 0.39 ^B^	11.75 ± 0.86 ^A^
**3**	8.00 ± 0.01 ^D^	25.00 ± 0.20 ^B^	10.09 ± 0.73 ^A^
**4**	8.06 ± 0.01 ^C^	25.89 ± 1.05 ^AB^	12.43 ± 0.50 ^A^
**5**	8.99 ± 0.02 ^B^	25.95 ± 0.18 ^AB^	11.40 ± 0.45 ^A^
**MRS**	9.43 ± 0.01 ^A^	26.81 ± 4.86 ^A^	11.47 ± 1.29 ^A^

Note: Values are mean ± standard error of the mean (SEM), *n* = 3. Mean ± SEM within the same column that share similar superscript (A–D) are not significantly different (*p* > 0.05).

**Table 5 molecules-25-00779-t005:** Cell population, tryptophan production and serine consumption of *P. acidilactici* TP-6 for different media formulation in the steepest ascent experiment.

Run	Cell Population (log CFU/mL)	Tryptophan Production (mg/L)	Serine Consumption (mg/L)
**1**	9.10 ± 0.02 ^D^	27.73 ± 0.04 ^G^	8.72 ± 0.21 ^D^
**2**	9.10 ± 0.01 ^D^	35.79 ± 0.28 ^F^	9.05 ± 0.35 ^DE^
**3**	9.14 ± 0.01 ^CD^	42.64 ± 0.40 ^E^	9.24 ± 0.39 ^DE^
**4**	9.15 ± 0.01 ^C^	55.54 ± 0.36 ^C^	8.93 ± 0.01 ^DE^
**5**	9.24 ± 0.01 ^B^	69.05 ± 0.55 ^A^	9.53 ± 0.09 ^DEF^
**6**	9.24 ± 0.01 ^B^	59.84 ± 0.68 ^B^	10.15 ± 0.12 ^F^
**7**	9.25 ± 0.01 ^B^	51.67 ± 0.92 ^D^	9.65 ± 0.21 ^EF^
**8**	9.18 ± 0.01 ^C^	43.77 ± 0.23 ^E^	9.00 ± 0.11 ^DE^
**9**	9.16 ± 0.01 ^C^	36.74 ± 0.91 ^F^	7.56 ± 0.16 ^C^
**10**	9.04 ± 0.01 ^E^	21.98 ± 0.38 ^H^	4.21 ± 0.10 ^A^
**11**	8.77 ± 0.03 ^F^	10.02 ± 0.07 ^I^	5.11 ± 0.45 ^B^
**MRS**	9.37 ± 0.01 ^A^	27.69 ± 0.15 ^G^	7.69 ± 0.37 ^C^

Note: Values are mean ± standard error of the mean (SEM), *n* = 3. Mean ± SEM within the same column that share similar superscript (A–I) are not significantly different (*p* > 0.05).

**Table 6 molecules-25-00779-t006:** Central Composite Design (CCD) matrix with coded value and their corresponding experimental and predicted tryptophan production by *P. acidilactici* TP-6.

Std Run	A	B	C	D	Tryptophan Production (mg/L)	
					Experimental	Predicted *
1	−1	−1	−1	−1	56.24 ± 0.04 ^JKL^	56.86
2	1	−1	−1	−1	54.33 ± 0.20 ^N^	54.44
3	−1	1	−1	−1	53.23 ± 0.14 ^O^	53.54
4	1	1	−1	−1	54.49 ± 0.26 ^N^	54.72
5	−1	−1	1	−1	60.62 ± 0.17 ^G^	60.38
6	1	−1	1	−1	57.12 ± 0.23 ^IJK^	57.64
7	−1	1	1	−1	59.14 ± 0.32 ^H^	59.38
8	1	1	1	−1	59.69 ± 0.26 ^GH^	60.24
9	−1	−1	−1	1	63.25 ± 0.35 ^E^	63.20
10	1	−1	−1	1	56.32 ± 0.44 ^JKL^	56.78
11	−1	1	−1	1	59.25 ± 0.25 ^H^	59.44
12	1	1	−1	1	55.89 ± 0.85 ^LM^	56.62
13	−1	−1	1	1	64.47 ± 0.40 ^D^	64.92
14	1	−1	1	1	58.01 ± 0.56 ^I^	58.18
15	−1	1	1	1	63.13 ± 0.68 ^EF^	63.48
16	1	1	1	1	60.25 ± 0.44 ^GH^	60.34
17	−2	0	0	0	62.23 ± 0.29 ^EF^	61.88
18	2	0	0	0	57.19 ± 0.33 ^IJ^	56.32
19	0	−2	0	0	56.83 ± 0.12 ^IJKL^	56.36
20	0	2	0	0	55.95 ± 0.13 ^KLM^	55.20
21	0	0	−2	0	55.03 ± 0.22 ^MN^	54.28
22	0	0	2	0	61.99 ± 0.42 ^F^	61.52
23	0	0	0	−2	59.66 ± 0.27 ^GH^	59.04
24	0	0	0	2	66.07 ± 0.05 ^C^	65.48
25	0	0	0	0	69.33 ± 0.10 ^AB^	69.42
26	0	0	0	0	69.55 ± 0.31 ^AB^	69.42
27	0	0	0	0	70.22 ± 0.07 ^A^	69.42
28	0	0	0	0	68.85 ± 0.39 ^B^	69.42
29	0	0	0	0	69.69 ± 0.55 ^AB^	69.42
30	0	0	0	0	68.88 ± 0.94 ^B^	69.42
MRS	-	-	-	-	28.18 ± 0.12 ^P^	-

Note: Values are mean ± standard error of mean (SEM), *n* = 3. Mean ± SEM within the same column that share similar superscript (A–P) are not significantly different (*p* > 0.05). * Predicted tryptophan production was calculated based on Equation (3).

**Table 7 molecules-25-00779-t007:** ANOVA of regression model for tryptophan production by *P. acidilactici* TP-6.

Source	Sequential	Lack of Fit	Adjusted	Predicted	
	*p*-Value	*p*-Value	R-Squared	R-Squared	
Linear	0.1492	<0.0001	0.1059	0.0576	
Crossproduct	0.9724	<0.0001	−0.1063	−0.3333	
Quadratic	<0.0001	0.2196	0.9837	0.9586	Suggested
Cubic	0.9875	0.0265	0.9709	0.2223	Aliased

**Table 8 molecules-25-00779-t008:** ANOVA for quadratic model of tryptophan production by *P. acidilactici* TP-6.

Source	Sum of Squares	df	Mean Square	F-Value	*p*-value Prob > F	
Model	814.96	14	58.21	126.07	< 0.01	significant
A	46.16	1	46.16	99.97	< 0.01	significant
B	2.08	1	2.08	4.51	0.05	
C	78.3	1	78.3	169.58	< 0.01	significant
D	61.84	1	61.84	133.92	< 0.01	significant
AB	12.86	1	12.86	27.84	< 0.01	significant
AC	0.11	1	0.11	0.24	0.63	
AD	16.07	1	16.07	34.8	< 0.01	significant
BC	5.36	1	5.36	11.62	<0.01	significant
BD	0.2	1	0.2	0.42	0.53	
CD	3.17	1	3.17	6.86	0.02	significant
A^2^	182.37	1	182.37	394.96	< 0.01	significant
B^2^	318.65	1	318.65	690.1	< 0.01	significant
C^2^	227.15	1	227.15	491.94	< 0.01	significant
D^2^	87.9	1	87.9	190.37	< 0.01	significant
Residual	6.93	15	0.46			
Lack of Fit	5.57	10	0.56	2.06	0.22	not significant
Pure Error	1.35	5	0.27			
Cor Total	821.88	29				

Note: R^2^: 0.9916; Adj R^2^: 0.9837; Pred R^2^: 0.9586; Adeq Precision: 33.028.

**Table 9 molecules-25-00779-t009:** Coded and real values of variables selected in PBD for tryptophan production by *P. acidilactici* TP-6.

Variables	Symbol Code	Unit	Coded Values	
			−1	+1
**Glucose**	A	g/L	0	20
**Sucrose**	B	g/L	0	17.69
**Fructose**	C	g/L	0	19.08
**Lactose**	D	g/L	0	18.86
**Molasses**	E	g/L	0	25.08
**Yeast extract**	F	g/L	0	4
**Peptone**	G	g/L	0	10
**Meat extract**	H	g/L	0	8
**K_2_HPO_4_**	J	g/L	0	2
**KH_2_PO_4_**	K	g/L	0	2
**Urea**	L	g/L	0	3
**NH_4_NO_3_**	M	g/L	0	5
**(NH_4_)_2_SO_4_**	N	g/L	0	5
**(NH_4_)_2_HC_6_H_5_O_7_**	O	g/L	0	2
**NaOAc**	P	g/L	0	5
**MgSO_4_**	Q	g/L	0	0.2
**MnSO_4_**	R	g/L	0	0.04
**Tween 80**	S	mL/L	0	1
**FeSO_4_**	T	g/L	0	0.01
**ZnSO_4_**	U	g/L	0	0.01
**CuSO_4_**	V	g/L	0	0.01
**Biotin**	W	g/L	0	0.06

**Table 10 molecules-25-00779-t010:** PBD matrix for 22 variables with coded values for tryptophan production by *P. acidilactici* TP-6.

Std Run	A	B	C	D	E	F	G	H	J	K	L	M	N	O	P	Q	R	S	T	U	V	W	X
1	1	1	1	1	1	−1	1	−1	1	1	−1	−1	1	1	−1	−1	1	−1	1	−1	−1	−1	−1
2	−1	1	1	1	1	1	−1	1	−1	1	1	−1	−1	1	1	−1	−1	1	−1	1	−1	−1	−1
3	−1	−1	1	1	1	1	1	−1	1	−1	1	1	−1	−1	1	1	−1	−1	1	−1	1	−1	−1
4	−1	−1	−1	1	1	1	1	1	−1	1	−1	1	1	−1	−1	1	1	−1	−1	1	−1	1	−1
5	−1	−1	−1	−1	1	1	1	1	1	−1	1	−1	1	1	−1	−1	1	1	−1	−1	1	−1	1
6	1	−1	−1	−1	−1	1	1	1	1	1	−1	1	−1	1	1	−1	−1	1	1	−1	−1	1	−1
7	−1	1	−1	−1	−1	−1	1	1	1	1	1	−1	1	−1	1	1	−1	−1	1	1	−1	−1	1
8	1	−1	1	−1	−1	−1	−1	1	1	1	1	1	−1	1	−1	1	1	−1	−1	1	1	−1	−1
9	−1	1	−1	1	−1	−1	−1	−1	1	1	1	1	1	−1	1	−1	1	1	−1	−1	1	1	−1
10	−1	−1	1	−1	1	−1	−1	−1	−1	1	1	1	1	1	−1	1	−1	1	1	−1	−1	1	1
11	1	−1	−1	1	−1	1	−1	−1	−1	−1	1	1	1	1	1	−1	1	−1	1	1	−1	−1	1
12	1	1	−1	−1	1	−1	1	−1	−1	−1	−1	1	1	1	1	1	−1	1	−1	1	1	−1	−1
13	−1	1	1	−1	−1	1	−1	1	−1	−1	−1	−1	1	1	1	1	1	−1	1	−1	1	1	−1
14	−1	−1	1	1	−1	−1	1	−1	1	−1	−1	−1	−1	1	1	1	1	1	−1	1	−1	1	1
15	1	−1	−1	1	1	−1	−1	1	−1	1	−1	−1	−1	−1	1	1	1	1	1	−1	1	−1	1
16	1	1	−1	−1	1	1	−1	−1	1	−1	1	−1	−1	−1	−1	1	1	1	1	1	−1	1	−1
17	−1	1	1	−1	−1	1	1	−1	−1	1	−1	1	−1	−1	−1	−1	1	1	1	1	1	−1	1
18	1	−1	1	1	−1	−1	1	1	−1	−1	1	−1	1	−1	−1	−1	−1	1	1	1	1	1	−1
19	−1	1	−1	1	1	−1	−1	1	1	−1	−1	1	−1	1	−1	−1	−1	−1	1	1	1	1	1
20	1	−1	1	−1	1	1	−1	−1	1	1	−1	−1	1	−1	1	−1	−1	−1	−1	1	1	1	1
21	1	1	−1	1	−1	1	1	−1	−1	1	1	−1	−1	1	−1	1	−1	−1	−1	−1	1	1	1
22	1	1	1	−1	1	−1	1	1	−1	−1	1	1	−1	−1	1	−1	1	−1	−1	−1	−1	1	1
23	1	1	1	1	−1	1	−1	1	1	−1	−1	1	1	−1	−1	1	−1	1	−1	−1	−1	−1	1
24	−1	−1	−1	−1	−1	−1	−1	−1	−1	−1	−1	−1	−1	−1	−1	−1	−1	−1	−1	−1	−1	−1	−1

**Table 11 molecules-25-00779-t011:** Media formulation to validate the effects of significant variables on tryptophan production by *P. acidilactici* TP−6.

Media Formulation	Medium Composition, g/L
**Medium 1**	
Glucose	20
Sucrose	17.69
Lactose	18.86
Molasses	25.08
Yeast extract	4
Peptone	10
Meat extract	8
K_2_HPO_4_	2
Urea	3
(NH_4_)_2_SO_4_	5
(NH_4_)_2_HC_6_H_5_O_7_	2
NaOAc	5
MgSO_4_	0.2
MnSO_4_	0.04
FeSO_4_	0.01
CuSO_4_	0.01
**Medium 2**	
Meat extract	8
FeSO_4_	0.01
Lactose	18.86
MnSO_4_	0.04
Urea	3
CuSO_4_	0.01
K_2_HPO_4_	2
MgSO_4_	0.2
**Medium 3**	
Meat extract	8
FeSO_4_	0.01
Lactose	18.86
MnSO_4_	0.04
Urea	3
CuSO_4_	0.01
K_2_HPO_4_	2
MgSO_4_	0.2
Biotin	0.06
Tween 80	1
**Medium 4**	
Lactose	18.86
Meat extract	8
Urea	3
FeSO_4_	0.01
**Medium 5**	
Molasses	25.08
Meat extract	8
Urea	3
FeSO_4_	0.01

**Table 12 molecules-25-00779-t012:** Steepest Ascent design for tryptophan production by *P. acidilactici* TP-6.

No.	Run	Variable level, g/L			
		Molasses (A)	Meat Extract (B)	Urea (C)	FeSO_4_ (D)
	Δ	−2.51	4	0.6	0.003
1	Origin	25.08	8	3.0	0.010
2	Origin + Δ	22.57	12	3.6	0.013
3	Origin + 2Δ	20.06	16	4.2	0.016
4	Origin + 3Δ	17.55	20	4.8	0.019
5	Origin + 4Δ	15.04	24	5.4	0.022
6	Origin + 5Δ	12.53	28	6.0	0.025
7	Origin + 6Δ	10.02	32	6.6	0.028
8	Origin + 7Δ	7.51	36	7.2	0.031
9	Origin + 8Δ	5.00	40	7.8	0.034
10	Origin + 9Δ	2.49	44	8.4	0.037
11	Origin + 10Δ	0	48	9.0	0.040

**Table 13 molecules-25-00779-t013:** Coded and real values of variables selected for CCD of tryptophan production by *P. acidilactici* TP-6.

Variables	Coded Symbol	Coded Values				
		−α	−1	0	+1	+α
Molasses	A	10.02	12.53	15.04	17.55	20.06
Meat extract	B	16	20	24	28	32
Urea	C	4.2	4.8	5.4	6	6.6
FeSO_4_	D	0.016	0.019	0.022	0.025	0.028

**Table 14 molecules-25-00779-t014:** CCD matrix for four variables with coded values for tryptophan production by *P. acidilactici* TP-6.

Std Run	A	B	C	D
1	−1	−1	−1	−1
2	1	−1	−1	−1
3	−1	1	−1	−1
4	1	1	−1	−1
5	−1	−1	1	−1
6	1	−1	1	−1
7	−1	1	1	−1
8	1	1	1	−1
9	−1	−1	−1	1
10	1	−1	−1	1
11	−1	1	−1	1
12	1	1	−1	1
13	−1	−1	1	1
14	1	−1	1	1
15	−1	1	1	1
16	1	1	1	1
17	−2	0	0	0
18	2	0	0	0
19	0	−2	0	0
20	0	2	0	0
21	0	0	−2	0
22	0	0	2	0
23	0	0	0	−2
24	0	0	0	2
25	0	0	0	0
26	0	0	0	0
27	0	0	0	0
28	0	0	0	0
29	0	0	0	0
30	0	0	0	0

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
