# Peer review of "Rapid Evaluation and Optimization of Medium Components Governing Tryptophan Production by Pediococcus acidilactici TP-6 Isolated from Malaysian Food via Statistical Approaches"

_molecules, 2020, doi:10.3390/molecules25040779_

Round 1

Reviewer 1 Report

In this manuscript, the production of tryptophan by a lactic acid bacteria P. acidilactici TP-6 was optimized by via Plackett-Burman Design and Central Composite Design. The unpathogenic microorganism that produced tryptophan have great application potential in feed and food industries. The manuscript is technically sound. The optimization process for tryptophan production was detailed and effective. In general, the manuscript is acceptable for publication in this journal with minor revision.

There are some observations or recommendations listed as followed:

The authors mentioned the result of the the highest predicted tryptophan production and validation experiment in the Abstract and Conclusion sections. But they were missing in the Result section. Please supplement these results.

The authors should compare their tryptophan production with other reports in the result section for better understanding their advantages.

How did the authors design the medium formula used in Table 4? In Table 4, the authors should indicated the medium formula was listed in Table 11.

Author Response

Comments and Suggestions for Authors

In this manuscript, the production of tryptophan by a lactic acid bacteria P. acidilactici TP-6 was optimized by via Plackett-Burman Design and Central Composite Design. The unpathogenic microorganism that produced tryptophan has great application potential in feed and food industries. The manuscript is technically sound. The optimization process for tryptophan production was detailed and effective. In general, the manuscript is acceptable for publication in this journal with minor revision.

There are some observations or recommendations listed as followed:

Point 1: The authors mentioned the result of the highest predicted tryptophan production and validation experiment in the Abstract and Conclusion sections. But they were missing in the Result section. Please supplement these results.

Response 1: Results for the highest predicted tryptophan production and validation experiment were included in the Results & Discussion section at Line 561-583, which was highlighted in cyan box.

Point 2: The authors should compare their tryptophan production with other reports in the result section for better understanding their advantages.

Response 2: We have compared the tryptophan production with other reports accordingly in the Results section at Line 561-583, which was highlighted in cyan box.

Point 3: How did the authors design the medium formula used in Table 4? In Table 4, the authors should indicate the medium formula was listed in Table 11.

Response 3: We have indicated the medium formula in Table 4 was listed in Table 11 at Line 374, which was highlighted in cyan box.

Reviewer 2 Report

The study presents the research results on ‘Rapid Evaluation and Optimization of Medium 2 Components Governing Tryptophan Production’. The topic and content of the paper seem appropriate for 'Molecules'. However, adding in-depth discussions that reflect recent research trends will help readers understand the paper.

A concise abstract is required. Also, more specific descriptions of authors' finding should be added in the abstract rather than overall result of study. The abstract should state briefly the purpose of the research, the principal results and major conclusions.

There are many research results about tryptophan production. Please provide the advantage of this study compare to other recent research papers regarding the productivity. What is the excellence of this paper in terms of productivity?

The explanation of the results from the statistical approach seems appropriate. It seems necessary to add an in-depth discussion of the results. In particular, it would be easier for readers to understand if a description of the correlation between variables is added.

To reflect recent research trends, please add information about ‘Tryptophan Productivity’ in the Results and discussion section.

Citations from some references do not seem appropriate. Authors need to add papers published in 2017-2019 that reflect recent research trends.

Author Response

Comments and Suggestions for Authors

The study presents the research results on ‘Rapid Evaluation and Optimization of Medium 2 Components Governing Tryptophan Production’. The topic and content of the paper seem appropriate for 'Molecules'.

Point 1: However, adding in-depth discussions that reflect recent research trends will help readers understand the paper.

Response 1: The current manuscript is one of the pioneering works focusing on exploitation of lactic acid bacteria for the production of amino acid. Hence, there were very limited publications related to the present work. However, we have included several related recent publications pertaining to the exploration of LAB for amino acid production, particularly recent references [11], [12], [25] and [61] that are closely related.

Point 2: A concise abstract is required. Also, more specific descriptions of authors' finding should be added in the abstract rather than overall result of study. The abstract should state briefly the purpose of the research, the principal results and major conclusions.

Response 2: We have revised the abstract according to the comments of reviewer 2 to make it more concise and highlighted the feasibility of rapid optimisation of medium components through statistical approaches.

Point 3: There are many research results about tryptophan production. Please provide the advantage of this study compare to other recent research papers regarding the productivity. What is the excellence of this paper in terms of productivity?

Response 3: We have compared the tryptophan production with other reports accordingly in the Result section at Line 561-583, which was highlighted in cyan box.

Point 4: The explanation of the results from the statistical approach seems appropriate. It seems necessary to add an in-depth discussion of the results. In particular, it would be easier for readers to understand if a description of the correlation between variables is added.

Response 4: The correlations between variables at Line 484-559 was described in detail for all response changes pertaining to the interactive effects between 2 variables at a time.

Point 5: To reflect recent research trends, please add information about ‘Tryptophan Productivity’ in the Results and discussion section.

Response 5: We have included the tryptophan productivity of the producer strain in the Results and discussion section accordingly at Line 561-583, which was highlighted in cyan box.

Point 6: Citations from some references do not seem appropriate. Authors need to add papers published in 2017-2019 that reflect recent research trends. 

Response 6: The current manuscript is one of the pioneering works focusing on the exploitation of lactic acid bacteria for amino acid production. Hence there are very limited articles related to the present work and we have included several closely related recent publications ([11], [12], [25] and [61]) pertaining to the exploration of LAB for amino acid production in current report.

Round 2

Reviewer 2 Report

Authors have provided reasonable response to this reviewer's comments.

Editor's Point:

Point 2: It is very important for you to give a detail explanation for novelty of your manuscript since we have found much repetitions with one of your previous work in Microbial cell factories, 2019, 18(1), 125.

Response:

For optimisation studies via statistically approach, first order design such as Plackett Burman Design is normally employed to screen for the significant variables, followed by a second order design such as Central Composite Design.

Therefore, the works reported in this manuscript was performed by using similar approaches as our previous work published in Microbial cell factories entitled “Optimized Medium via Statistical Approach Enhanced Threonine Production by Pediococcus pentosaceus TL-3 Isolated from Malaysian Food”.
The previous work was focused on the production of threonine by P. pentosaceus TL-3, whereas the present work described the production of tryptophan by P. acidilactici TP-6. Hence, the producer strains used in both studies were different for 2 different optimisation studies for the corresponding threonine and tryptophan amino acid productions.

Furthermore, the amino acid produced in both studies belongs to different families, which involves different biosynthetic pathways. For instance, threonine belongs to the aspartate family that utilises aspartate as a precursor. In contrast, tryptophan production occurs via shikimate pathway, initiated by the condensation of phosphoenolpyruvate and erythrose-4-phosphate.

Therefore, both works that reported in the current manuscript  and our previous published work in Microbial cell factories, 2019, 18(1), 125 are very novel. To our knowledge, both are the first report for the optimisation study of medium components for threonine and tryptophan amino acid productions by 2 different Lactic acid bacteria of P. pentosaceus TL-3 and P. acidilactici TP-6 respectively that isolated from Malaysian foods. The results presented in both reports have proven further the optimisation strategy via statistical approaches employed in both reports is an effective and feasible
approach that can be used for the optimisation of medium components for amino acid productions by lactic acid bacteria.